# Does the “Morning Morality Effect” Apply to Prehospital Anaesthesiologists? An Investigation into Diurnal Changes in Ethical Behaviour

**DOI:** 10.3390/healthcare8020101

**Published:** 2020-04-16

**Authors:** Anne Craveiro Brøchner, Lars Grassmé Binderup, Caroline Schaffalitzky de Muckadell, Søren Mikkelsen

**Affiliations:** 1Department of Anaesthesiology and Intensive Care Medicine V, Mobile Emergency Care Unit, Odense University Hospital, 5000 Odense, Denmark; Anne.Craveiro.Broechner@rsyd.dk; 2Department of Regional Health Research, University of Southern Denmark, 5000 Odense, Denmark; 3Department of Anaesthesiology, Kolding Hospital, a Part of Hospital Lillebaelt, 6000 Kolding, Denmark; 4Philosophy, Department for the Study of Culture, University of Southern Denmark, 5230 Odense, Denmark; binderup@sdu.dk (L.G.B.); csm@sdu.dk (C.S.d.M.)

**Keywords:** morning morality, ethical correct behaviour, admission pattern

## Abstract

The “morning morality effect”—the alleged phenomenon that people are more likely to act in unethical ways in the afternoon when they are tired and have less self-control than in the morning—may well be expected to influence prehospital anaesthesiologist manning mobile emergency care units (MECUs). The working conditions of these units routinely entail fatigue, hunger, sleep deprivation and other physical or emotional conditions that might make prehospital units predisposed to exhibit the “morning morality effect”. We investigated whether this is in fact the case by looking at the distribution of patient transports to hospital with and without physician escort late at night at the end of the shift as a surrogate marker for changing thresholds in ethical behaviour. All missions over a period of 11 years in the MECU in Odense were reviewed. Physician-escorted transports to hospital were compared with non-physician-escorted transports during daytime, evening, and night-time (which correlates with time on the 24 h shifts). In total, 26,883 patients were transported to hospital following treatment by the MECU. Of these, 27.4% (26.9%–27.9%) were escorted to the hospital. The ratio of patient transports to hospital with and without physician escort during the three periods of the day did not differ (*p* = 1.00). We found no evidence of changes in admission patterns over the day. Thus, no evidence of the expected “morning morality effect” could be found in a prehospital physician-manned emergency care unit.

## 1. Introduction

This study investigates the moral conduct of anaesthesiologist working in rapid-response mobile emergency care units (MECUs). In general, evidence has suggested that people may be less likely to behave unethically in the morning than in the afternoon. This phenomenon has been termed “morning morality effect” [1] and has been observed in judges, who have been shown to let legally irrelevant situational determinants, for example having a food break, influence their rulings [2]. In broader terms, this phenomenon may result in humans being more prone to unethical behaviour at some points of the day than others [1]. This and other kinds of biases in ethical judgements have attracted a lot of interest in research in recent years, as findings suggest that decisions are often highly influenced by non-rational factors such as hunger, disgust, or personal interest [3,4].

Whether the “morning morality effect” is relevant in relation to the behaviour of physicians has not yet been investigated. If, however, such behaviour does exist in the medical field, a likely place to find it would be among prehospital physicians. First, prehospital physicians may be summoned late at night regardless of sleep and food deprivation. This in itself has been reported to negatively influence their performance and the safety of the patients [5]. Second, a new mission may be ordered just before the termination of a shift, causing the physicians to have to work overtime regardless of the preceding workload. Third, the prehospital anaesthesiologist usually acts as the sole physician at the prehospital scene without the possibility of peer consultation, feedback, or supervision. Fourth, the diagnostic or interventional decisions required prehospitally often need to be taken quickly with little time for deliberation and are usually based on no or very little background information. All of these external factors may lead to an increased risk of a less than morally optimal performance at the scene during odd hours.

These working conditions apply to the anaesthesiologist-manned prehospital MECUs in Odense, Denmark. The MECU shifts are 24 h on weekdays and 12 h on weekends, starting at 7:30 a.m. or p.m. but always ending at 7:30 a.m. The MECU is dispatched by an emergency dispatcher based on specific criteria. This dispatch is not influenced by the anaesthesiologist manning the MECU [6]. Following contact with the patient, three mission outcomes are possible, i.e., the patient can be released at the scene after treatment, the patient can be escorted to the hospital by the anaesthesiologist, or the patient can be transported to the hospital in the ambulance without escort by the anaesthesiologist. These mission outcomes are decided solely by the physician present at the scene. There are many situations in which there is clearly no reason for a physician escort to the hospital. In contrast, a patient in cardiac arrest or an intubated patient being treated for septic shock obviously should be escorted by the physician during transport to the hospital. However, grey areas do exist in which a given patient would have been escorted to the hospital by one prehospital physician, while another prehospital physician might have found it less burdensome to hand over the responsibility of the patient care to the ambulance personnel and subsequently return to the base.

The basis for our study was the assumption that handing over a patient to the ambulance personnel to perform an un-escorted transport of the patient to the hospital might be considered a less burdensome task for the anaesthesiologist than to physically escort the patient to the hospital in the ambulance. If behavioural traits akin to the “morning morality effect” were to occur, we speculated that this could be revealed in the form of differing patterns of escorting the patient to the hospital depending on the time of the day and shift. Given that we had previously found that the documentation of ethical considerations during resuscitation was almost absent [7] and given that the conditions that might lead to the “morning morality effect” are present in the MECUs, we decided to investigate whether such unethical behaviour or ethical depletion could be detected in a prehospital MECU late at night.

## 2. Methods

### 2.1. Study Setting and Population

This study was carried out in the MECU in Odense in the Region of Southern Denmark. The MECU in Odense is part of the nationally implemented and publicly funded three-tiered emergency medical system in which the basic resource is an ambulance manned by two emergency medical technicians. The prehospital resource is dispatched by an emergency medical dispatch centre where the dispatcher, a health care worker, dispatches an ambulance, an ambulance and a paramedic, or an ambulance and a physician-manned rapid-response vehicle [6]. The MECU in Odense is an anaesthesiologist-manned rapid-response vehicle whose team consists of an anaesthesiologist and an emergency medical technician with special training assisting the anaesthesiologist. In this particular MECU, each anaesthesiologist has a prehospital work experience ranging from 3 years to more than 20 years. The MECU operates 24/7 all year round. The length of the shifts is 24 h on weekdays and 12 h on weekends. The shifts terminate at 7:30 a.m. (and 7:30 p.m. on weekends). The prehospital anaesthesiologists all work under a general contract applicable to all MECUs in the region. The contractual fixed salary for the prehospital anaesthesiologist does not depend on the number of patients treated or the outcome of the missions. The average workload on working days amounts to 8 h from 16:00 to 7:30 the next morning. The average workload during weekends amounts to 8 h per shift. There are no chores between missions, and the MECU physician may relax or take a nap when possible. The MECU in Odense covers approximately 2500 square km while servicing a population of 260,000 people. Almost all patients admitted to the emergency department or the hospital following treatment by the MECU are transported to the Odense University Hospital, the only tertiary hospital in the region. On average, the MECU is dispatched in 26.2% of all ambulance runs in the region, corresponding to approximately 13 missions per day [8]. After concluding each mission, the MECU physician enters system variables and patient variables in a register that was established 1 October 2007, remaining unchanged through the whole observation period. Among other variables, of relevance for this study, the prehospital anaesthesiologist registers the type of mission according to the following classification: Patient transported to hospital by ambulance escorted by the anaesthesiologist; Patient transported to hospital by ambulance without escort by the anaesthesiologist; Patient released at the scene following treatment; Patient declared dead without reliable signs of death; Patient declared dead exhibiting reliable signs of death; Mission down-prioritised after patient contact in favour of another mission; Mission down-prioritised before patient contact in favour of another mission; MECU waived by ambulance; Stand-by missions (fires, police actions); Patient not found; Miscellaneous.

This registration is mandatory, and the physician cannot terminate the shift before the mission outcome is registered for all missions during the shift.

### 2.2. Study Design

The study was a register-based retrospective study in which the mission outcome of all MECU missions carried out from 1 October 2007 until 31 December 2018 were analysed.

The number of patients transported to the hospital with physician escort was compared with the number of patients transported to the hospital without physician escort by stratifying the time of day into three strata: daytime (8:00–15:59), evening (16:00–23:59), night (00:00–07:59).

### 2.3. Statistical Analyses

Data are presented as proportions or median and quartiles (where appropriate). Proportions are presented with 95% confidence intervals (CI) based on a binomial distribution. Chi-squared tests were applied to categorical data presented in cross tables. Furthermore, we analysed the data using logistic regression analysis with “admission without escort by the anaesthesiologist” and “admission escorted by the anaesthesiologist” as the dependent dichotomous variable and working day and weekend and time of day (day, evening, night) as independent variables. We further performed logistic regression analysis with “admission without escort by the anaesthesiologist” and “admission escorted by the anaesthesiologist” as the dependent dichotomous variable, adjusting for day of the week, National Advisory Committee for Aeronautics score [9], and prehospital diagnosis group as assigned by the MECU physician [10]. Differences were considered significant when *p* < 0.05. All statistical calculations were performed using STATA 15.1 (StataCorp, College Station, TX, USA).

### 2.4. Legislative Approval of the Study

As no personally identifiable information regarding patients or the involved physicians was extracted from the registry, according to the Danish legislation, no approvals from the Ethics Board, the Danish Data Protection Agency, or the Danish Health and Medicines Authorities were required for this study.

## 3. Results

During an observation period of 11 years and three months, the MECU was dispatched a total of 48,272 times, resulting in encounters with 37,366 patients. A total of 26,883 of these patients were transported to hospital following treatment by the MECU. For the distribution of the missions, see Figure 1.

During the observation period of more than 11 years, we found a steadily decreasing number of MECU missions. While the proportion of patients that were escorted to hospital by the MECU rose in that period, the total number of patients transported to hospital by the MECU declined (see Table 1).

For 20 patients, information regarding the time of day was missing. Of the remaining 26,863 patients with complete data, a total of 7356 patients were escorted to the hospital by the physician. The remaining 19,507 patients were transported to hospital without physician escort. The daytime and the evening were the busiest periods, with 11,241 and 10,254 patients transported to hospital following treatment. During the night-time, the number of patients admitted to hospital following treatment was 5368. The relative distribution of patients transported to hospital with and without physician escort during the three periods of the day did not differ (*p* = 1.00) (see Table 2).

We found a significant difference in the physician transport pattern investigated by the hour. A slight increase in the fraction of patients escorted to hospital was thus observed between 3:00 a.m. and 5:00 a.m. (Table 3, Figure 2).

Logistic regression analysis with “admission without escort by the anaesthesiologist” and “admission escorted by the anaesthesiologist” as the dependent binary variable and weekdays or weekends and time of day as the independent variable did not show any difference throughout the observation period, as the odds ratio for escorted transport to hospital did not change significantly at any time (See Table 4).

A logistic regression analysis with “admission without escort by the anaesthesiologist” and “admission escorted by the anaesthesiologist” as the dependent binary variable and time of day as the independent variable, adjusted for diagnosis groups, day of the week, and NACA score was inconclusive (data not shown).

The distribution of patients within the International Statistical Classification of Diseases and Related Health Problems 10th Revision (ICD-10) diagnosis groups [10] is shown in Table 4. Patients with infections and circulatory and respiratory diseases and pregnant women and patients with injuries were predominant in the group of physician-escorted patients (Table 5).

## 4. Discussion

The principles of the Hippocratic Oath should govern decisions made by a physician in any patient encounter [11,12], and all physicians should act conscientiously and in a morally correct way whatever the time of day and regardless of circumstances. [11] In our investigation, we looked for evidence of a lower quality in ethical judgements late at night in the MECU when the anaesthesiologist must decide whether or not to escort a patient from the scene of a prehospital incident to the hospital. We did not find any such evidence.

This is a welcomed conclusion in a situation where claims and complaints against doctors are growing worldwide [13] and cases have been reported of physicians’ unethical behaviour towards patients [10] and of physicians’ bias towards patients based on personal characteristics [14,15]. There have also been cases where unsound associations with pharmaceutical companies have led to biased decisions made by the physicians, adversely impacting patient outcomes or wastefully increasing health care costs [16,17]. There have also been cases of fraud in medical science with potentially far-reaching consequences [18,19,20] and reports of unethical behaviour in relation to the use of social media among surgical residents and faculty surgeons [21].

In that context, the Hippocratic Oath is arguably as relevant today as it was 2400 years ago [22,23], as are the four basic principles of ethical conduct towards patients drafted in 1994 [24]. These principles provide a shared ethical framework, upon which all physician–patient encounters can be based regardless of religion and which are neutral among competing political, cultural, and philosophical theories [24]. However, the publication of rules guiding ethical behaviour cannot solve all problems. We have to remain vigilant and to look out for potential unethical practise. There is no reason to believe that physicians conduct themselves at a higher moral level than non-physicians, and given the large number of physicians worldwide, it is not surprising to find cases of physicians slacking on the job, becoming corrupt, losing abilities and skills, or exploiting their professional positions for personal gain [25]. This study focuses on one potential, more subtle kind of un-ethical behavior among physicians—the alleged phenomenon of “morning morality”, where ethical behaviour declines in the course of the day.

It has generally been suggested that there may be time slots during a shift when the individual, due to fatigue, makes less correct and less timely decisions. The topic of decision-making in such settings has previously been sporadically investigated in relation to emergency medical technicians (EMTs) but not in relation to physicians. In EMTs, tiredness has, for instance, indeed been reported to influence decision-making regarding the transportation of patients [26] and, when approaching the end of a shift, has been reported to be a factor influencing the intensity of the treatment or the decision to treat at the scene or to transport to the hospital [27]. In a field closely related to prehospital work, it has been shown that the duration of the average emergency medical centre dispatcher call was affected by night work shift, increasing by about 10 s late at night [28]. Differences have likewise been reported in outcome over the course of the day for anaesthetic adverse events, death in the Intensive Care Unit, dialysis care, and operation start time [29].

It is thus well documented that fatigue among health care workers induced by long shifts can lead to adverse results. One suggested cause of such adverse outcome is that fatigue may impair cognitive abilities (e.g., attention span, memory, ability to reason reliably) that affect the quality of clinical judgment. Here, we rather focused on a related, yet distinct, potential cause of adverse outcome, namely, cases where late-shift fatigue influences the physician’s ability to weigh personal interests against the obligation to care for the patient in an ethically responsible way.

In short, we hypothesised that a constant exertion of self-control during the workday may lead to a state of ego depletion, reducing people’s moral identity and increasing the individual´s propensity to engage in unethical behaviour [30]. The capacity for self-control (which in this context is understood as the ability to make the right—but potentially inconvenient—decisions) has been investigated in previous research [31,32]. According to the strength model of self-regulation, the capacity for self-control requires rest after use. All acts of self-control thus draw from the same finite resource, and the depletion of that resource hinders a person’s ability to subsequently exert self-control [31,32]. This finding is supported by reports that dishonesty increases when people’s capacity to exert self-control is impaired [33].

In the case of prehospital physician, the circumstances under which decisions are typically made arguably create a heightened risk of allowing morally irrelevant determinants to influence ethical judgement. This is because the prehospital physician is acting as the sole decision-maker at the prehospital scene and has to act under time pressure and with little background information. Although shared decision-making may consider the opinions of the patient into consideration regarding admission to hospital or not [34], formally and legally, the authority to decide the transport mode rests with the physician present at the scene. These conditions are in stark contrast with the conditions that apply to critical decision-making within hospitals. In hospital, decisions are most often made following discussions with multiple peers, and this plenary-type consultation arguably has a generally positive influence on the quality of moral judgements [35,36]. Prehospital settings lack similar peer discussions because in most instances, only one prehospital physician and one to three EMTs are present. These conditions deprive the physician the opportunity to confer with colleagues concerning specific cases, treatment strategies, or even transportation strategies. More importantly, under these circumstances, the physician is not under the obligation to justify decisions explicitly to fellow physicians, which might otherwise have had a mitigating effect on the influence of irrelevant self-interest on judgement. In short, the lack of peer discussion and of demand for accountability may make the prehospital anaesthesiologist more likely to make a decision that is influenced by lack of sleep, disturbances of REM sleep, or considerations regarding the end of a shift, when choosing whether to escort a patient to the hospital or simply to return to the base for rest or replacement.

These considerations have led us to investigate a potential risk of unethical behaviour in prehospital settings in the form of “morning morality effect”, ethical depletion, or ethically inappropriate slacking. Little is known about whether this is merely a theoretical concern. However, since differences have been reported in outcome over the course of the day for anaesthetic adverse events, death in the ICU, dialysis care, and operation start time [29], investigations are called for.

It is important to emphasise that we are not necessarily hypothesising that the circumstances at the prehospital scene cause a *deliberate* lowering of the ethical thresholds by the individual physician. Rather, it is reasonable to expect that exhaustion experienced during the late hours of a shift [3,37,38] may change an individual’s alertness and inadvertently allow morally irrelevant determinants to influence ethical judgement. We believe that—were such a phenomenon to be documented—it would be unethical not to take systemic countermeasures.

Recognizing these potential obstacles to ethically correct decision-making in the prehospital setting, we investigated whether the admission pattern of a MECU in the odd hours of a shift differed from that during the rest of the day. Specifically, we used the mission outcome “transported to hospital unescorted by an anaesthesiologist” as a surrogate marker for a possible lowered quality of ethical judgements. The rationale for choosing “transport without escort by an anaesthesiologist” as a marker for lowered quality of ethical judgements is that the decision to escort a patient to the hospital is a more demanding and time-consuming task than merely letting the ambulance personnel transport the patient to the hospital without escort by an anaesthesiologist. The MECU is organized in 24 h shifts on weekdays and 12 h shifts on weekends (always ending at 7:30 a.m.), which means that the early morning are the odd hours at the end of each shift. Using differences in the pattern of admissions of patients during the hours of the day as a surrogate marker for a possible lowered quality of ethical judgements, corresponding to the “morning morality effect”, we found no evidence of change in behaviour during the hours of the day.

### 4.1. Strengths of the Study

This study covers a period of more than 11 years, in which a total of 26,883 patients were transported to hospital. A total of 18 different anaesthesiologists were employed at the MECU during that period. We believe that reporting results from a diverse group of physicians covering a large span of years and a large number of patient encounters adds validity to our study. Furthermore, the study is a retrospective study. We have thus dispensed with any potential Hawthorne effect [39]. We believe that this also increases the validity of our study.

### 4.2. Limitations of the Study

The effect parameter “non-physician escorted hospital admission” may be insufficient as a surrogate parameter for unethical behaviour related to the time of day, and our study is susceptible to arguments that we have simply investigated “slacking on the job” or indolence. Gunia et al. have thus claimed that the compatibility between a person’s chronotype (the propensity for the individual to sleep at a particular time during a 24 h period) and the time of day offers a more complete predictor of that person’s ethicality than does the time of day alone [40].

As such, the prerequisite for accepting the validity of our research question is that the mission outcome “physician escort” can be accepted as a surrogate parameter for ethically correct behaviour. We contend that not all patients transported to hospital should be escorted by the prehospital physician. However, the fraction of patients being escorted to hospital should probably not diminish at night, and had we found that this fraction had indeed diminished at night, this finding would have been problematic.

One thus could argue that in order to firmly ascertain whether the prehospital behavior of the anaesthesiologist was ethically correct, each mission should have been evaluated on an individual basis. With other 20,000 missions completed within the observation period, this would have been an impossible task.

A further limitation of the study is that the diagnoses assigned to patients that were escorted to the hospital differed from the diagnoses assigned to patients not escorted to the hospital. It should, however, come as no surprise that some patients (for example patients with circulatory or respiratory diseases) were predominantly escorted to the hospital. It has been shown previously that patients in these two diagnosis groups have a higher mortality rate than other patients [8]. One may thus assume that the degree of severity of illness was greater in these patients than in patients in the other ICD-10 classification system groups.

Furthermore, the workload of the MECU varied throughout the years. During the observation period of more than 11 years, we found a steadily decreasing number of MECU missions. While the proportion of patients that were escorted to hospital by the MECU rose in that period, the total number of patients transported to hospital by the MECU declined. This trend goes against the trend that the prehospital services in a Danish health region increased from 24.3 missions per 1000 inhabitants in 2007 to 40.2 missions per 1000 inhabitants in 2014 [41]. However, during the observation period of 11 years, the general educational level of the EMTs and the paramedics has increased in the Region of Southern Denmark. This may have had implications for the level of severity of illness that the ambulances could handle without assistance from the MECU and, as such, may have reduced the need for assistance from the MECU in cases of lesser severity. The increasing competences of the EMTs correspond to a lowered number of missions and a higher number of patients escorted to the hospital by the anaesthesiologist manning the MECU. These organisational changes, however, should have no impact on the diurnal variation in admission that we report. One substantial weakness of our study is that what should have been an obviously preferable statistical analysis of our data, a logistical regression analysis, was probably not appropriate, based on the very low values for McFadden R^2^. The models did not fit, given the available data. We were thus limited to the application of more basic statistical tools to evaluate the data. We believe, however, that because of our large observation period and our rather extensive data pool, our findings do suggest that possible unethical variation, “morning morality effect”, or slacking in the quality of decision-making is not an issue in the anaesthesiologist-manned MECU.

## 5. Conclusions

If the theory of “morning morality” is correct and it is true that it is more burdensome for prehospital anaesthesiologists to physically escort a patient to the hospital in the ambulance than not to, then we would expect to observe a change in the admission pattern over the course of the day and shift. This model of investigating unethical traits in the treatment of patients is admittedly simple and it does not directly provide us with information concerning concrete moral judgements. We do, however, consider the results to be relevant and illuminating, because the working conditions of the prehospital anaesthesiologists may be suspected of leading him or her towards unethical decision-making late at night. In this study, we found no evidence of unethical behavioural traits associated with “morning morality”. This is reassuring, as diurnal variations in vigilance have been found in other personnel groups working late at night [5,42].

## Figures and Tables

**Figure 1 healthcare-08-00101-f001:**
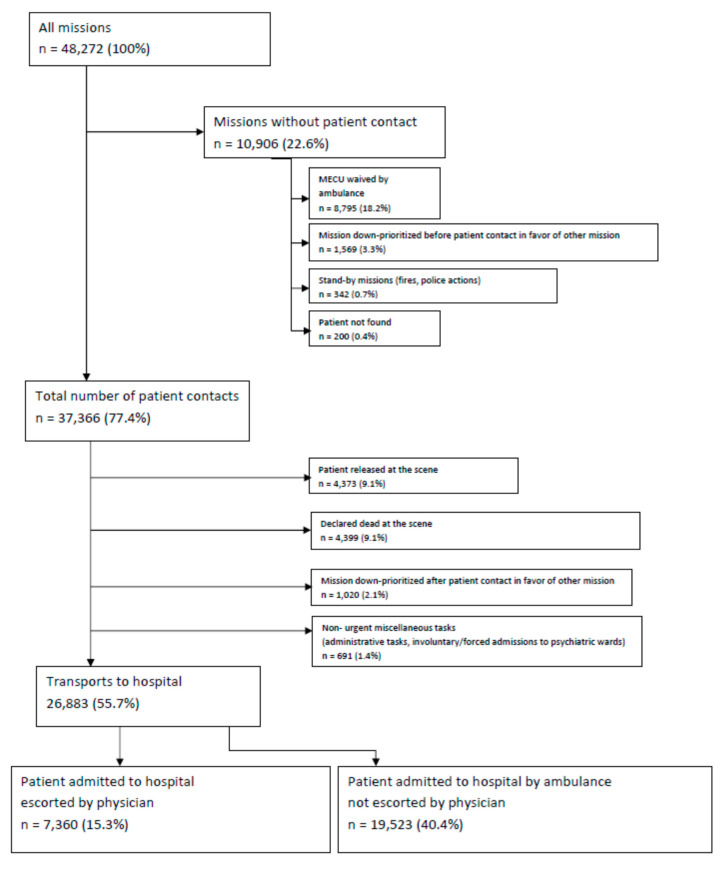
Flowchart depicting the distribution of mobile emergency care unit (MECU) missions in 2007–2018.

**Figure 2 healthcare-08-00101-f002:**
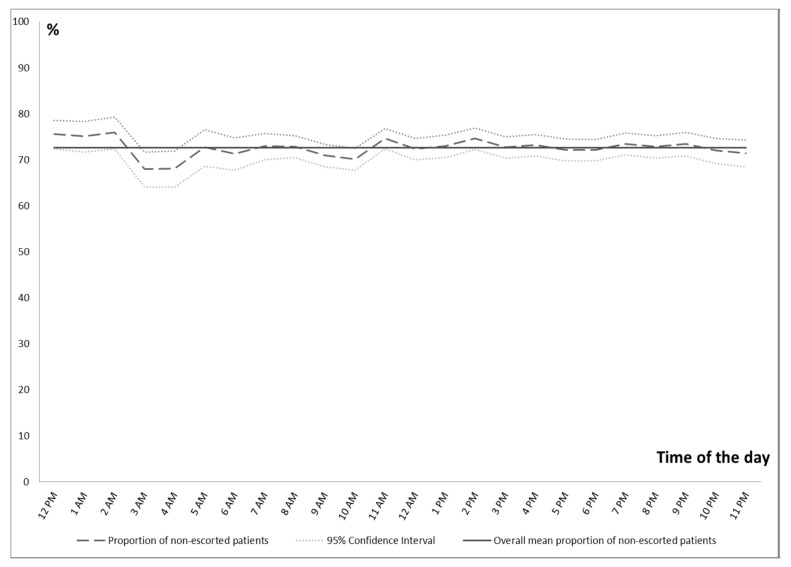
Proportion of transported patients not escorted to hospital. (Dashed line: proportion of non-escorted patients; dotted lines: 95% confidence intervals; solid line: mean proportion of non-escorted patients).

**Table 1 healthcare-08-00101-t001:** Development in the number of transported patients over the years 2007–2018. Data presented as numbers (percentages (95% CI)).

Year	Transported to Hospital with Escort *n* (%(95%CI))	Transported to Hospital without Escort *n* (%(95%CI))	Total no. of Patients Transported to Hospital
2007 (4th quarter)	229 (27.9 (24.8–31.1))	593 (72.1 (68.9–75.2))	822
2008	725 (22.9 (21.5–24.4))	2439 (77.1 (75.6–78.5))	3164
2009	683 (22.0 (20.6–23.5))	2416 (78.0 (76.5–79.4))	3099
2010	619 (22.3 (20.8–23.9))	2151 (76.7 (76.1–79.2))	2770
2011	639 (24.0 (22.4–25.7))	2021 (76.0 (74.3–77.6))	2660
2012	596 (23.2 (21.6–24.9))	1974 (76.8 (75.1–78.4))	2570
2013	598 (25.5 (23.7–27.3))	1750 (74.5 (72.7–76.3))	2348
2014	685 (29.5 (27.7–31.4))	1635 (70.5 (68.6–72.3))	2320
2015	664 (34.6 (32.4–36.7))	1257 (65.4 (63.3–67.6))	1921
2016	628 (34.9 (32.7–37.1))	1173 (65.1 (62.9–67.3))	1801
2017	610 (38.3 (35.9–40.7))	983 (61.7 (59.3–64.1))	1593
2018	684 (37.7 (35.5–40.0))	1131 (62.3 (60.0–64.5))	1815

**Table 2 healthcare-08-00101-t002:** Modes of admission of patients transported to hospital with and without physician escort according to daytime shift, evening shift, and night-time shift.

	Day	Evening	Night	*p*–Value
Escorted by physician	3077	2811	1468	1.00
Unescorted by physician	8164	7443	3900	

**Table 3 healthcare-08-00101-t003:** Proportion of patients transported to hospital in relation to the time of the day. (Chi^2^; *p* < 0.035).

Time Interval	Total Number of Patients Admitted to Hospital	Patients Admitted to Hospital without Physician Escort (*n* (% (95%CI))
12:00 a.m.–0:59 a.m.	806	609 (75.5 (72.4–78.5))
1:00 a.m.–1:59 a.m.	661	496 (75.0 (71.6–78.3))
2:00 a.m.–2:59 a.m.	612	465 (75.9 (72.4–79.3))
3:00 a.m.–3:59 a.m.	583	396 (67.9 (64.0–71.7))
4:00 a.m.–4:59 a.m.	551	375 (68.1 (64.0–71.9))
5:00 a.m.–5:59 a.m.	501	364 (72.7 (68.5–76.5))
6:00 a.m.–6:59 a.m.	672	479 (71.3 (67.7–74.7))
7:00 a.m.–7:59 a.m.	982	716 (72.9 (70.0–75.7))
8:00 a.m.–8:59 a.m.	1345	980 (72.9 (70.4–75.2))
9:00 a.m.–9:59 a.m.	1370	971 (70.9 (68.4–73.3))
10:00 a.m.–10:59 a.m.	1439	1009 (70.1 (67.7–72.5))
11:00 a.m.–11:59 a.m.	1450	1082 (74.6 (72.3–76.8))
12:00 p.m.–0:59 p.m.	1473	1066 (72.4 (70.0–74.6))
1:00 p.m.–1:59 p.m.	1358	991 (73.0 (70.5–75.3))
2:00 p.m.–2:59 p.m.	1360	1014 (74.6 (72.2–76.9))
3:00 p.m.–3:59 p.m.	1446	1051 (72.7 (70.3–75.0))
4:00 p.m.–4:59 p.m.	1459	1067 (73.1 (70.8–75.4))
5:00 p.m.–5:59 p.m.	1381	996 (72.1 (69.7–74.5))
6:00 p.m.–6:59 p.m.	1469	1059 (72.1 (69.7–74.4))
7:00 p.m.–7:59 p.m.	1351	992 (73.4 (71.0–75.8))
8:00 p.m.–8:59 p.m.	1327	966 (72.8 (70.3–75.2))
9:00 p.m.–9:59 p.m.	1197	879 (73.4 (70.8–75.9))
10:00 p.m.–10:59 p.m.	1127	811 (72.0 (69.2–74.6))
11:00 p.m.–11:59 p.m.	943	673 (71.4 (68.4–74.2))
Total	26,863	19,507 (72.6 (72.1–73.1))

**Table 4 healthcare-08-00101-t004:** Odds ratio for escorted transport to hospital, weekends vs. working days.

Characteristics	Odds Ratio	95% CI	*p*-Value
Day of week			
Sat.–Sun.	1 (Reference)		
Mon.–Fri.	1.02	0.96–1.08	0.61
Time of day			
08:00 a.m.–15:59 p.m.	1 (Reference)		
16:00 p.m.–23:59 p.m.	1.00	0.94–1.06	0.94
00:00 a.m.–07:59 a.m.	1.00	0.93–1.08	0.99

McFadden R^2^ = 0.0000.

**Table 5 healthcare-08-00101-t005:** Distribution of diagnoses for patients transported to hospital with and without physician escort.

ICD-10 Chapter	Diagnosis Group	Patients (Total)	Patients Escorted to Hospital by Physician (Proportion of Subgroup)		Patients not Escorted to Hospital by Physician (Proportion of Subgroup)	
			(*n*)	(% (95% CI))	(*n*)	(% (95% CI))
Chapter I	Certain infectious and parasitic diseases (A00–B99)	284	147	51.8 (45.8–57.7)	137	48.2 (42.3–54.2)
Chapter II	Neoplasms (C00–D48)					
Chapter III	Diseases of the blood and blood-forming organs and certain disorders involving the immune mechanism (D50–D89)					
Chapter IV	Endocrine, nutritional and metabolic diseases (E00–E90)	413	61	14.8 (11.5–18.6)	352	85.2 (81.4–88.5)
Chapter V	Mental and behavioural disorders (F00–F99)	533	74	13.9 (11.1–17.1)	459	86.1 (82.9–88.9)
Chapter VI	Diseases of the nervous system (G00–G99)	658	168	25.5 (22.2–29.0)	490	74.5 (71.0–77.7)
Chapter VII	Diseases of the eye and adnexa (H00–H59)					
Chapter VIII	Diseases of the ear and mastoid process (H60–H95)					
Chapter IX	Diseases of the circulatory system (I00–I99)	5245	2252	42.9 (41.6–44.3)	2993	57.1 (55.7–58.4)
Chapter X	Diseases of the respiratory system (J00–J99)	3129	997	31.9 (30.2–33.5)	2132	68.1 (66.5–69.8)
Chapter XI	Diseases of the digestive system (K00–K93)	354	92	26.0 (21.5–30.9)	262	74.0 (69.1–78.5)
Chapter XII	Diseases of the skin and subcutaneous tissue (L00–L99)					
Chapter XIII	Diseases of the musculoskeletal system and connective tissue (M00–M99)	133	2	1.5 (0.2–5.3)	131	98.5 (94.7–99.8)
Chapter XIV	Diseases of the genitourinary system (N00–N99)	83	11	13.3 (6.8–22.5)	72	86.7 (77.5–93.2)
Chapter XV	Pregnancy, childbirth and the puerperium (O00–O99)	91	34	37.4 (27.4–48.1)	57	62.6 (31.9–72.6)
Chapter XVI	Certain conditions originating in the perinatal period (P00–P96)	11	2	18.1 (2.3–51.8)	9	81.8 (48.2–97.7)
Chapter XVII	Congenital malformations, deformations and chromosomal abnormalities (Q00–Q99)					
Chapter XVIII	Symptoms, signs and abnormal clinical and laboratory findings, not elsewhere classified (R00–R99)	4937	978	19.8 (18.7–20.9)	3959	80.2 (79.1–81.3)
Chapter XIX	Injury, poisoning and certain other consequences of external causes (S00–T98)	4042	1397	34.6 (33.1–36.1)	2645	65.4 (63.9–66.9)
Chapter XX	External causes of morbidity and mortality (V01–Y98)					
Chapter XXI	Factors influencing health status and contact with health services (Z00–Z99)	6851	1122	16.4 (15.5–17.3)	5729	83.6 (82.7–84.5)
Chapter XXII	Codes for special purposes (U00–U85)					
Missing		109	20	18.3 (11.6–26.9)	89	81.7 (73.1–88.4)
Total		26,883	7360	27.4 (26.9–27.9)	19,523	72.3 (72.1–73.2)

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
