# Peer review of "Does the “Morning Morality Effect” Apply to Prehospital Anaesthesiologists? An Investigation into Diurnal Changes in Ethical Behaviour"

_healthcare, 2020, doi:10.3390/healthcare8020101_

Round 1

Reviewer 1 Report

General comments:

The topic of this paper is interesting is innovative, by focusing on the study of a possible cognitive phenomena that may influence the ethical behaviour of physicians. In general, the Introduction would benefit from more detailed references in order to better clarify the rationale of the authors, as is done in the Discussion section. The methods are adequately described, and the study design is appropriate. Regarding the intelligibility of the language, only minor spell checking is required.

Specific comments:

Introduction

  • The rationale for the study is not very clear in a first moment, but it is possible to ascertain it from the Introduction section. However, I felt that the topic could be better justified by adding more literature references about how the “morning morality effect” can threaten work ethics and which are the mechanisms underlying it (which is better explained in the Discussion section).
  • This effect could also be linked to the vast existing literature in Psychology about cognitive biases and how they affect human reasoning.

Methods

  • The methods are explained in a clear way.
  • During all the years of data registration, were the data registered in the same form?
  • Please format the category answer options for the type of mission, for example, separating them by semicolons.

Discussion

  • Line 243: in this sentence, in which the authors are referring to literature from other authors can be misleading since it may be read as referring to the results of the present study (whose results do not support the occurrence of unethical behaviour in the form of “morning morality effect”.

Author Response

Rebuttal to reviewer one´s Specific comments:

Introduction

The rationale for the study is not very clear in a first moment, but it is possible to ascertain it from the Introduction section. However, I felt that the topic could be better justified by adding more literature references about how the “morning morality effect” can threaten work ethics and which are the mechanisms underlying it (which is better explained in the Discussion section).

Authors´response:

We have extended the introductions section hoping that we have been able to set the scene for the investigation somewhat more convincing.

This effect could also be linked to the vast existing literature in Psychology about cognitive biases and how they affect human reasoning.

Authors´ response:

We have expanded the section somewhat and entered a few more relevant references.

Methods

The methods are explained in a clear way.

During all the years of data registration, were the data registered in the same form?

Authors´ response:

This has been clarified in the text: “…register that was established 1st of October 2007, remaining unchanged through the whole observation period.”

Please format the category answer options for the type of mission, for example, separating them by semicolons.

Authors´ response:

Corrected.

Discussion

Line 243: in this sentence, in which the authors are referring to literature from other authors can be misleading since it may be read as referring to the results of the present study (whose results do not support the occurrence of unethical behaviour in the form of “morning morality effect”.

Authors´ response:

The sentence has been re-written and should now appear more intelligble.

Reviewer 2 Report

Classification of ethical and non ethical attitude of physician based only in the simple fact that patients were transported accompagnied or not by a physician is very disturbing. I totally agree that proprtion of cheating and unethical attitude in physician is probably similar to genaral population, however i  might agree that it would be influenced on during daytime ,however current study was flawly designed to detect it . 

In order to verify the hypothesis the authors should check the unethical behavior for every transport and then split day and night transport between unethical attitude.

Therefore I believe title should be changed appropriately and focus only on current finding which are the proprtion of patient transported or not during day and night and may be focus on each pathology. it might not represent the initial purpose or try and finf unethical behavior in a sample of patients for each period then compare it.

Author Response

Rebuttal to Comments and Suggestions for Authors

Classification of ethical and non ethical attitude of physician based only in the simple fact that patients were transported accompagnied or not by a physician is very disturbing. I totally agree that proprtion of cheating and unethical attitude in physician is probably similar to genaral population, however i  might agree that it would be influenced on during daytime ,however current study was flawly designed to detect it . 

Authors´ response:

We propose that the very prerequisite for accepting our arguments is that one may consider that for the physician, escorted transports are considered as being more burdensome than unescorted transports. This we state very clearly in our paper.

In order to verify the hypothesis the authors should check the unethical behavior for every transport and then split day and night transport between unethical attitude.

Authors´ response:

This is an impossible task. Checking each and every mission retrospectively for more than 11 years simply cannot be done – both for practical reasons and for ethical reasons. There is no way the Danish legislation will allow for scrutiny of medical journals from approximately 20.000 patients for that purpose. We have entered a sentence in the paper stating this problem as a limitation:

One thus could argue that in order to firmly ascertain whether the prehospital behavior of the anaesthesiologist was ethically correct, each mission should have been evaluated on an individual basis. With other 20.000 missions completed within the observation period, this would have been an impossible task.

Therefore I believe title should be changed appropriately and focus only on current finding which are the proprtion of patient transported or not during day and night and may be focus on each pathology. it might not represent the initial purpose or try and finf unethical behavior in a sample of patients for each period then compare it.

Authors´ response:

As we write above: The prerequisite for understanding this paper as an investigation into morning morality is the acceptance that the ratio of escorted versus non-escorted patients may be seen as an indicator of the general willingness of the physician to take on a more burdensome task late at night.